# Exosomes Derived from Human Umbilical Cord Mesenchymal Stem Cells Accelerate Diabetic Wound Healing via Promoting M2 Macrophage Polarization, Angiogenesis, and Collagen Deposition

**DOI:** 10.3390/ijms231810421

**Published:** 2022-09-09

**Authors:** Liping Teng, Maria Maqsood, Min Zhu, Yuting Zhou, Mingzhu Kang, Juan Zhou, Jinghua Chen

**Affiliations:** 1Wuxi School of Medicine, Jiangnan University, Wuxi 214122, China; 2School of Life Sciences and Heath Engineering, Jiangnan University, Wuxi 214122, China

**Keywords:** exosomes, human-umbilical-cord-derived mesenchymal stem cells, diabetes, wound healing, macrophage polarization

## Abstract

Some scholars have suggested that the clinical application of exosomes derived from human umbilical cord mesenchymal stem cells (hucMSCs-exo) might represent a novel strategy to improve diabetic wound healing. However, the mechanisms underlying the effects of hucMSCs-exo on wound healing remain poorly understood. This study aimed to identify the mechanism of hucMSCs-exo in treating diabetic wounds. HucMSCs-exo were isolated from human umbilical cord mesenchymal stem cells (hucMSCs) and subcutaneously injected into full-thickness wounds in diabetic rats. Wound healing closure rates and histological analysis were performed. The levels of tumor necrosis factor-α (TNF-α), macrophage mannose receptor (MMR/CD206), platelet endothelial cell adhesion molecule-1 (PECAM-1/CD31), and vascular endothelial growth factor (VEGF) were detected by immunohistochemistry. The degree of collagen deposition was examined using Masson’s trichrome staining. Gross evaluation of wound healing was carried out from day 0 to 14 post-surgery, and the wound site was harvested for histology on days 3, 7, and 14 post-wounding. HucMSCs-exo transplantation increased diabetic wound healing. In vitro, hucMSCs-exo promoted the proliferation of human umbilical vein endothelial cells (HUVECs) and NIH-3T3 cells. In vivo, hucMSCs-exo reduced wound area and inflammatory infiltration and increased collagen fibers. In addition, wound tissues in the hucMSCs-exo group had higher CD206, CD31, and VEGF expressions and lower TNF-α levels than those in the control group on day 14. Our results demonstrated that hucMSCs-exo facilitated diabetic wound repair by inducing anti-inflammatory macrophages and promoting angiogenesis and collagen deposition.

## 1. Introduction

Skin wounds, especially chronic wounds, bring a substantial burden to patients and society [1,2]. According to the causative etiologies, chronic wounds are generally classified into three main categories: diabetic ulcers, vascular ulcers, and pressure ulcers [3,4]. Diabetic wounds, as one of the major chronic wounds, represent a growing health problem, leading to a significant source of disability and mortality in diabetic patients. In China, nearly 8.1% of diabetic patients over 50 years old suffer new diabetic wound injury in just one year, which accounts for 40% of 5-year mortality of cases with amputation treatment [5]. In diabetic patients, the wound healing process is disturbed as a result of neuropathy and vascular dysfunction. Diabetic wounds are characterized by a chronic inflammatory response and angiogenic imbalance, leading to the suppression of proliferation and tissue remodeling [6]. These pathological disorders need urgent clinical treatments. Although different treatments have been made to cure diabetic wounds, the best treatment strategies are still being developed.

In the past years, stem cells have appeared as strong tools to cure chronic skin wounds with many diverse advantages such as easy accessibility and no harm to the donors [7,8,9,10]. Human-umbilical-cord-derived mesenchymal stem cells (hucMSCs) exhibit different superior biological characteristics that prove very useful in tissue regeneration and wound healing [11,12]. HucMSCs therapy is a potential treatment strategy for wound healing. However, the direct transplantation of hucMSCs remains limited because of many risk factors such as malignant teratoma formation and surplus immune responses in the body [13,14]. Exosomes are small-sized particles with sizes ranging from 30 to 150 nm, showing similar molecule characteristics to their parent cells. These small membrane particles play a crucial role in intracellular communication by delivering nucleic acids, proteins, and lipids to the target cells or remodeling the extracellular matrix. The direct use of exosomes derived from stem cells may not only play the same functions as stem cells, but also avoid adverse effects caused by stem-cell-based therapy [15,16]. Innovative studies highlighted the potential role of hucMSCs-derived exosomes (hucMSCs-exo) in regenerative medicine [17,18]. HucMSCs-exo showed less immunogenicity and was superior in bio-compatibility. Recent studies have confirmed the vital role of exosomes as an intracellular communication system in the regeneration of multiple tissues like skin, kidney, liver, and so on [19,20,21]. Recent reviews of the literature on exosomes derived from mesenchymal stem cells (MSCs-exosomes) in regenerative medicine have found that MSCs-exosomes have great potential for treating chronic wounds by promoting skin cell proliferation, angiogenesis, and wound healing [22,23]. In this regard, it has been suggested that the clinical application of hucMSCs-exo might represent a novel strategy to improve wound healing [24,25]. However, the mechanisms underlying these effects of hucMSCs-exo on wound healing remain poorly understood.

In this study, we hypothesized that hucMSCs-derived exosomes could promote chronic wound healing by activating the transition from an inflammatory phase to a proliferation phase during the wound healing process. The rate of wound closure, inflammation, angiogenesis, and collagen content were investigated in a diabetic wound model to identify the mechanism of hucMSCs-exo in treating chronic wounds. It was demonstrated that hucMSCs-exo facilitated chronic wound repair by inducing anti-inflammatory macrophages and promoting angiogenesis and collagen deposition.

## 2. Results

### 2.1. Characterization of hucMSCs-Exo

After the extraction of hucMSCs-exo from the culture supernatant of hucMSCs via ultracentrifugation, hucMSCs-exo was characterized by transmission electron microscope (TEM), Zetasizer Nano ZS, and Western blotting, as shown in Figure 1. Isolated hucMSCs-exo had a spherical shape with a diameter range of 60 to 150 nm (Figure 1A). Particle size analysis showed that the average diameter of hucMSCs-exo was approximately 96 nm in size (Figure 1B). In addition, the tetraspanins CD9, CD63, and tumor susceptibility gene 101 (TSG101) as exosomal markers were detected by Western blotting (Figure 1C), indicating that CD9, CD63, and TSG101 were enriched in isolated hucMSCs-exo.

### 2.2. Effects of hucMSCs-Exo on the Proliferation of HUVECs and NIH-3T3 Cells In Vitro

It is described in previous studies that the cell proliferation of skin tissues in chronic wounds is a vital process of wound healing. Fibroblasts are the major cell type responsible for wound healing. This process is attributed to the formation and remodeling of new blood capillaries and blood vessels. It was found that hucMSCs-exo at a concentration range of 10 to 50 µg/mL significantly increased the proliferation of human umbilical vein endothelial cells (HUVECs) on day 3 (Figure 2A). However, no such change was observed in the hucMSCs-exo group after 24 h. To investigate the effect of hucMSCs-exo on the activity of fibroblasts, the proliferation of mouse fibroblast NIH-3T3 cells was assessed by 3-(4,5-dimethylthiazol-2-yl)-2,5-diphenyl tetrazolium bromide (MTT) assay. As shown in Figure 2B, the results revealed that NIH-3T3 cells treated with hucMSCs-exo at the concentration of 20, 30, 40, and 50 µg/mL for 24 and 72 h presented a significantly higher proliferation rate, compared with the untreated control group. These findings demonstrated that the proliferation of vascular endothelial cells and fibroblasts was stimulated by hucMSCs-exo treatment in vitro.

### 2.3. Effects of hucMSCs-Exo on Wound Healing in Diabetic Rats

To investigate the wound healing role of hucMSCs-exo in the chronic diabetic wound, full-thickness dorsal wound surgery in streptozotocin (STZ)-induced diabetic Sprague–Dawley (SD) rats was performed. The wound area of diabetic rats treated with phosphate-buffered saline (PBS) or hucMSCs-exo for 0, 3, 7, 10, and 14 days was measured. As shown in Figure 3, compared with the control group, diabetic wound closure was significantly accelerated in the hucMSCs-exo-treated diabetic rats. There was an obvious difference in wound closure rate of hucMSCs-exo-treated diabetic rats in comparison with the control group treated by PBS. The results indicated that the wound closure rate was significantly increased at day 7 post-wounding in the hucMSCs-exo group as compared with the control group (*p* < 0.05). Thus, the wound size reduction was 34.8% for PBS and 83.6% for hucMSCs-exo within 7 days (Figure 3B). On day 14, the wound closure rate was 98.1% in rats exposed to hucMSCs-exo, indicating that the wound was approximately healed after being treated with hucMSCs-exo. However, the wound healing rate in the control group was 89.7% on day 14. These results suggested that hucMSCs-exo could noticeably expedite wound closure by activating wound healing phases.

To further analyze the healing effects of hucMSCs-exo on the diabetic wounds, histopathological changes were evaluated by hematoxylin and eosin (H&E) staining (Figure 4). According to the results of histopathological observations, hucMSCs-exo considerably decreased the number of inflammatory cells on day 7 in comparison with the control group. On day 14, there was still a large number of inflammatory cells at the wound sites treated with PBS, while no obvious infiltration of inflammatory cells was observed in diabetic wounds treated with hucMSCs-exo. HucMSCs-exo significantly inhibited chronic inflammation in diabetic wounds. It was demonstrated that hucMSCs-exo could quickly control the chronic inflammation in the diabetic wound, suggesting that the inflammatory phase of wound healing was shortened after hucMSCs-exo treatment. In addition, on day 7 post-wounding, there was a significant difference between the hucMSCs-exo group and the control group in new epithelial formation. Furthermore, hucMSCs-exo significantly increased the epithelial thickness on day 14, but there was no characteristic structure of the epithelium in the PBS-treated group. Similarly, compared with the control group, new blood vessels were significantly increased in the diabetic wounds exposed to hucMSCs-exo within two weeks, indicating that hucMSCs-exo was involved in enhanced neoangiogenesis.

### 2.4. Effects of hucMSCs-Exo on the Inflammation in Diabetic Wounds

To further observe the change in inflammation reaction in diabetic wounds treated with PBS and hucMSCs-exo, immunohistochemical analysis was performed to evaluate the expression of tumor necrosis factor-α (TNF-α) and macrophage mannose receptor (MMR/CD206) in diabetic wound tissues. As shown in Figure 5A,B, a lower level of TNF-α as an inflammation marker was observed in tissue sections with the treatment of hucMSCs-exo than that in the control group on day 3 and day 14. Meanwhile, the expression of CD206, as an anti-inflammatory M2 macrophage marker, was downregulated on day 3 when the diabetic wounds were treated with hucMSCs-exo (Figure 5C,D) compared with the treatment of PBS. In contrast, there was a higher expression of CD206 on day 14 in hucMSCs-exo treatment than that in the control group (*p* < 0.05). These data proved that hucMSCs-exo treatment was involved in the anti-inflammatory effect as a direct result of the inhibition of pro-inflammatory cytokines, such as TNF-α. Furthermore, the anti-inflammatory effect of hucMSCs-exo in diabetic wounds was weak on day 3 and obviously enhanced on day 14, owing to a relative increase in M2 macrophages.

### 2.5. Effects of hucMSCs-Exo on Angiogenesis in Diabetic Wounds

It was further explored whether the hucMSCs-exo enhanced the angiogenesis at the wound sites of diabetic rats by immunohistochemistry analysis for platelet endothelial cell adhesion molecule-1 (PECAM-1/CD31) and vascular endothelial growth factor (VEGF). As shown in Figure 6A,B, the expression of CD31 as a marker of neovascularization was higher in diabetic wounds treated with hucMSCs-exo than that in the untreated group. Especially, there was a significant difference in the number of new blood vessels on day 7 between the hucMSCs-exo group and the control group (*p* < 0.05). Immunohistochemistry staining for VEGF was also performed to detect vascular development and angiogenesis. As shown in Figure 6C,D, the expression level of VEGF gradually increased on days 3, 7, and 14 post-wounding in the hucMSCs-exo-treated group. Conversely, there was a decreasing expression of VEGF in the control group. The findings indicated that hucMSCs-exo accelerated wound healing in the middle and late stages by inducing angiogenesis.

### 2.6. Effects of hucMSCs-Exo on Collagen Synthesis in Diabetic Wounds

The effect of hucMSCs-exo on wound extracellular matrix production was measured by Masson staining. In Figure 7, huge numbers of collagen fibers were observed in the granulation tissue of wounds treated with hucMSCs-exo in comparison with those in the control group. According to Figure 7B, there was a 1.6-fold increase in collagen deposition on day 7 post-wounding in the hucMSCs-exo-treated group compared with those treated with PBS (*p* < 0.05). Similarly, on day 14, dense collagen was significantly deposited in the wounds exposed to hucMSCs-exo, indicating that hucMSCs-exo could promote collagen synthesis and skin regeneration during the wound healing process.

## 3. Discussion

The process of wound healing is extremely complex, and is generally divided into three phases including inflammation, proliferation, and remodeling. The primary goal of the inflammatory phase is to clear pathogens and help increase vascular permeability, mainly depending on the activation of neutrophils and macrophages [26]. The proliferative stage of wound healing comprises the expansion of the granulation tissue, re-epithelialization, and new vasculature formation through the activity of keratinocytes and fibroblasts [27]. During the final stage, adequate collagen from type III to type I is deposited in an orderly way, leading to tissue remodeling and wound repair. However, the process of diabetic wound healing is slower than that of normal wound healing because of several bioactive factors including elevated pro-inflammatory factors and decreased growth factors [28]. Exosomes are considered good mediators for intercellular communication and the regulation of physiological and pathological processes. Recent reviews indicated that stem-cell-derived extracellular vesicles or exosomes had potential therapeutics for wound healing [29,30]. In this study, exosomes derived from hucMSCs were prepared and applied to the full-thickness wounds in diabetic rats, leading to the acceleration of diabetic wound healing. HucMSCs-exo reduced more wound area than the control treatment. According to our results, hucMSCs-exo exhibited a higher wound closure rate, indicating that exosomes transmitted information to the extracellular microenvironment or recipient cells [31]. Piyush Gondaliya et al. demonstrated that mesenchymal-stem-cell-derived exosomes loaded with miR-155 inhibitor could ameliorate diabetic wound healing [32]. Accumulation of evidence implied that hucMSCs-exo cargo might be responsible for the healing processes of diabetic wounds [33].

Under a high glucose environment, it is well known that the inflammatory phase is prolonged at wound sites, owing to an inhibition of macrophage polarization [34]. Macrophages as pioneer immune cells play a distinctive and vital role in the different stages of wound repair. Pro-inflammatory M1 macrophages can promote inflammation via the secretion of TNF-α and other inflammatory factors, while M2 phenotypes can elicit anti-inflammatory activities [35]. However, excessive inflammation would lead to impaired wound healing. In the present study, we found that transplanting hucMSCs-exo onto diabetic wounds effectively reduced the infiltration of inflammatory cells. Meanwhile, the expression of TNF-α decreased in the presence of hucMSCs-exo treatment, indicating the active role of hucMSCs-exo in anti-inflammatory effects. Furthermore, our results demonstrated that the amount of CD206, as a marker of M2 macrophages, decreased gradually in the control group, along with the relative increase in the hucMSCs-exo group. These findings are consistent with previous studies, which confirmed that exosomes from mesenchymal stem cells could induce the polarization of M1 to M2 macrophages [36,37].

Besides the inhibition of inflammation, optimization of vascular endothelial cells, fibroblasts, and keratinocytes is involved in wound closure [36]. Our findings confirmed that hucMSCs-exo had a positive effect on the proliferation of fibroblast NIH-3T3 cells and vascular endothelial cells in vitro, implying that hucMSCs-exo could play critical roles in ECM granulation tissue formation [27]. Although hucMSCs-exo markedly promoted the proliferation of HUVECs cells in vitro, we observed that hucMSCs-exo did not exert a significant influence on the expression of CD31 and VEGF in vivo compared with the control group. Hu et al. demonstrated that exosomes derived from bone-marrow-derived mesenchymal stem cells promoted VEGF and CD31 expression, leading to adequate angiogenesis in diabetic wound healing [38]. In addition, exosomes secreted by human-adipose-derived mesenchymal stem cells appeared more effective in treating cutaneous wounds, with the increased gene expression of VEGF [39]. In contrast to the previous findings stimulated by at least 100 μg of exosomes, our results, induced by only 10 μg of exosomes per wound, showed a trend of the relative increase in the expression of CD31 and VEGF at the middle and late stages, resulting in enhanced angiogenesis in H&E-stained tissues. Therefore, it seems that the effect of hucMSCs-exo on cell proliferation was more obvious than that on angiogenesis, owing to the differences in exosome cargo among exosomes derived from different mesenchymal stem cells [40].

Moreover, collagen deposition in an orderly manner is a key step in wound healing [28,41]. A scaffold for the deposition of collagen is formed by granulation tissue, leading to matrix remodeling and better healing outcomes [28]. A previous study demonstrated that adipose stem cell-derived microvesicles significantly increased re-epithelialization and collagen deposition to accelerate wound closure via AKT and extracellular signal-regulated kinase (ERK) signaling pathways [42]. A recent study has reported that exosomes from human-adipose-derived mesenchymal stem cells had wound-healing-promoting effects by enhancing collagen synthesis [43]. In agreement with the previous findings, our study confirmed that diabetic wounds treated by hucMSCs-exo led to more abundant collagen fibers than the control treatment. With the increase in healing time, a significantly higher density of collagens was observed in the hucMSCs-exo group. Thus, hucMSCs-exo is also involved in diabetic wound healing by increased collagen synthesis and deposition at the wound sites.

## 4. Materials and Methods

### 4.1. Materials

Dulbecco’s modified Eagle medium (DMEM, low-glucose) and fetal bovine serum (FBS) were purchased from Gibco, USA. An enhanced BCA protein assay kit was obtained from Shanghai Beyotime Biotechnology Co., Ltd., Shanghai, China. Antibodies against CD9, CD63, TSG101, TNF-α, CD206, CD31, and VEGF were obtained from Abcam, Cambridge, UK. 3-(4,5-dimethylthiazol-2-yl)-2,5-diphenyl tetrazolium bromide (MTT) reagent was purchased from Sangon Biotech Co., Ltd., Shanghai, China. Human umbilical cord mesenchymal stem cells (hucMSCs) were gifted by the Yangtze Delta Region Institute of Tsinghua University, Hangzhou public translational platform for cell therapy, China. Human umbilical vein endothelial cells (HUVECs) and mouse fibroblast NIH-3T3 cells were obtained from the Cell Bank of the Chinese Academy of Sciences, Shanghai, China. Sprague–Dawley (SD) rats (male, 6 weeks, 200 ± 20 g) were purchased from Shanghai SLAC Laboratory Animal Co., Ltd., Shanghai, China.

### 4.2. Isolation and Characterization of Exosomes from hucMSCs

HucMSCs were cultured in a low-glucose DMEM medium containing 10% exosome-depleted FBS. The culture supernatant of hucMSCs was collected to harvest hucMSCs-exo. HucMSCs-exo was extracted via a centrifugation and ultracentrifugation process, as described in previous protocols. Firstly, the supernatant was centrifuged at 300× *g* for 10 min to remove cell pellets. In the next step, the supernatant was centrifuged for 10 min at 2000× *g* and the pellets containing dead cells were removed. After removing cell debris at 10,000× *g* for 30 min, the supernatant was further ultracentrifuged at 100,000× *g* for 70 min. The supernatant was discarded, and then the pelleted exosomes were washed with a large volume of PBS and ultracentrifuged at 100,000× *g* for 70 min. Finally, exosomes were resuspended in 200 µL PBS and stored at −80 °C for further experiment.

The morphology of hucMSCs-exo was determined with an acceleration voltage of 120 kV using a transmission electron microscope (TEM). The diameter distribution of hucMSCs-exo was investigated by Zetasizer Nano ZS. To explore the identification of exosomes, exosome markers including CD9, CD63, and TSG101 were examined by Western blotting, as described previously.

### 4.3. Cell Proliferation Assay

To determine the effect of hucMSCs-exo on the proliferation of HUVECs and NIH-3T3 cells, an MTT assay was performed to detect cell viability. Briefly, HUVECs and NIH-3T3 cells were seeded in 96-well plates. After overnight incubation, the cell medium was taken out and adherent cells were treated with hucMSCs-exo at different concentrations. After 24 h or 72 h, the resulting formazan was dissolved with dimethyl sulfoxide (DMSO) and the absorbance of the solution was measured at 570 nm.

### 4.4. Preparation of Diabetic Rat Wound Models and Treatment

All procedures were approved by the Animal Ethics Committee of Jiangnan University, Wuxi, China. After SD rats were fed adaptively, 10 mg/mL streptozotocin (STZ) was intraperitoneally injected into SD rats at a dosage of 70 mg/kg. Rats with a blood glucose level ≥16.7 mmol/mL were considered to be diabetic and were employed in subsequent experiments.

After diabetic rats were anesthetized with 2% isoflurane, a full-thickness excisional wound (10 mm in diameter) was crafted on the dorsal skin of each rat. Then, the rats were randomly divided into two treatment groups, which were subcutaneously injected with 100 μL PBS (control group) and 100 μL hucMSCs-exo (100 μg/mL), respectively, around the wound at four injection sites on day 0. Overall health and behavioral changes of rats were closely scrutinized from day 0 to 14 post-surgery. Digital photographs were taken every day after treatment, while a ruler was placed adjacent to a wound model as a guide. The wound area was measured and the wound healing level was calculated by Image J on days 0, 3, 7, 10, and 14 post-wounding. Finally, the animals were euthanized on days 3, 7, and 14 and the tissues around the wound were excised for further assessment.

### 4.5. Histological Analysis

The excised tissues from wound sites were fixed in 4% paraformaldehyde for 24 h and then gradually dehydrated with graded ethanol. After harvested tissues were embedded in paraffin, the samples were sliced into a 4 μm thick paraffin section. Hematoxylin and eosin (H&E) staining was carried out to observe the histological changes in the wound healing process. Masson’s trichrome staining assay was performed to determine the degree of collagen maturity according to the manufacturer’s instructions. Images were captured under a microscope and quantified with the help of Image J software.

### 4.6. Immunohistochemistry Analysis

Immunohistochemistry analysis was used to determine the levels of TNF-α and CD206 as markers of pro/anti-inflammation in the diabetic wound tissues treated with hucMSCs-exo. CD31 and VEGF were detected by immunohistochemistry to study angiogenesis induced by hucMSCs-exo during the wound healing process. After skin sections were dewaxed in xylene and rehydrated, the antigen retrieval was performed by boiling in sodium citrate buffer (pH 6.0). Subsequently, tissue sections were incubated with different primary antibodies overnight at 4 °C and then treated with horseradish peroxidase-conjugated secondary antibody for 1 h. Finally, the samples were visualized using a DAB immunohistochemistry colour development kit. The images of the stained sections were obtained under a microscope.

### 4.7. Statistical Analysis

All experiments were performed with at least three replicates per group and the experiments were repeated at least three times. The Shapiro–Wilk test was performed to examine the normality of all variables. Normally distributed data were shown as means ± standard deviation (SD). The independent sample *t*-test was used to compare the differences between the two groups. Statistical analysis was conducted using 23.0 software. *p* < 0.05 was considered statistically significant.

## 5. Conclusions

In conclusion, our results showed that hucMSCs-exo played an important role in diabetic wound healing. HucMSCs-exo could obviously accelerate wound repair in diabetic rats by promoting M2 macrophage polarization, angiogenesis, and collagen deposition. In terms of clinical application, hucMSCs-exo might be applied to diabetic wounds with one-step in situ injection in the future. However, the precise underlying mechanism of hucMSCs-exo in diabetic wound treatment should be further elucidated, providing an effective therapeutic approach for diabetic wounds.

## Figures and Tables

**Figure 1 ijms-23-10421-f001:**
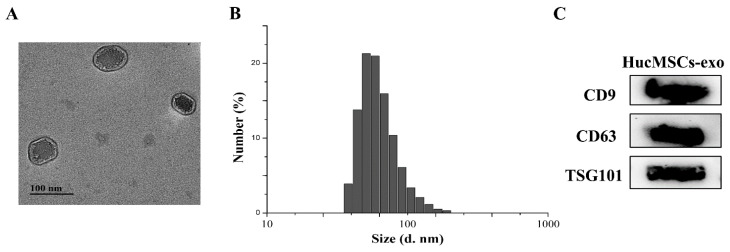
Characterization of hucMSCs-exo. (**A**) TEM micrograph of hucMSCs-exo. The scale bar is 100 nm. (**B**) Particle size and distribution analysis of hucMSCs-exo by Zetasizer Nano ZS. (**C**) Western blotting analysis of hucMSCs-exo markers.

**Figure 2 ijms-23-10421-f002:**
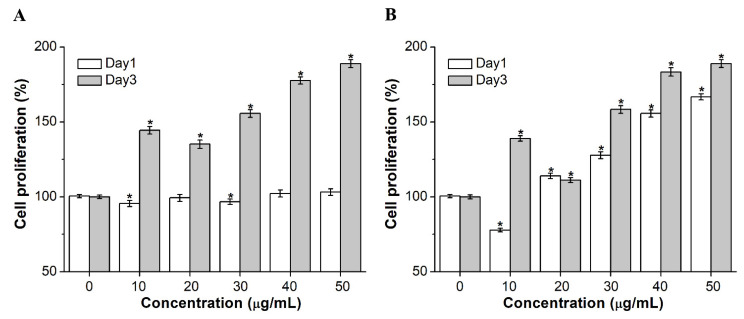
Proliferative effects of hucMSCs-exo on HUVECs and NIH-3T3 cells. (**A**) HUVECs and (**B**) NIH-3T3 cells were treated with hucMSCs-exo at different concentrations. Untreated hucMSCs-exo was used as the control group. Data represent the mean ± SD (n = 3), * *p* < 0.05.

**Figure 3 ijms-23-10421-f003:**
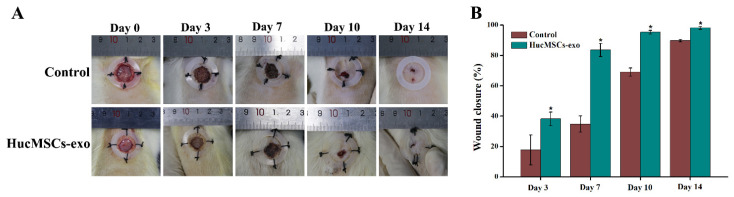
Diabetic wound healing induced by hucMSCs-exo in vivo. (**A**) Representative images of full-thickness dorsal wounds in STZ-induced diabetic rats at days 0, 3, 7, 10, and 14 post-wounding after treatment with phosphate-buffered saline (PBS as a control group) and hucMSCs-exo. (**B**) Quantification of wound closure at different time points in two groups (n = 5). * *p* < 0.05 shows a significant difference compared with the control group on the same day.

**Figure 4 ijms-23-10421-f004:**
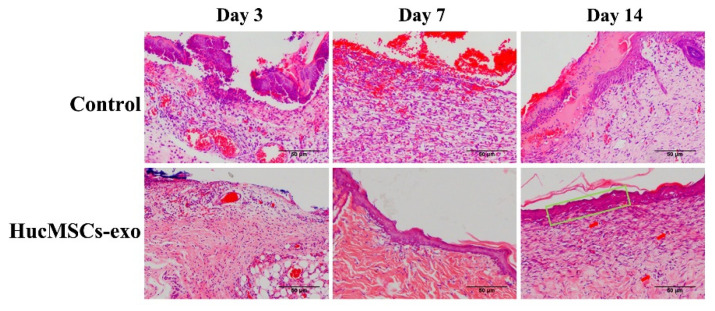
H&E staining of diabetic wound tissues treated with PBS (control) and hucMSCs-exo on days 3, 7, and 14. Red arrow: blood vessels, green rectangle: characteristic epithelium.

**Figure 5 ijms-23-10421-f005:**
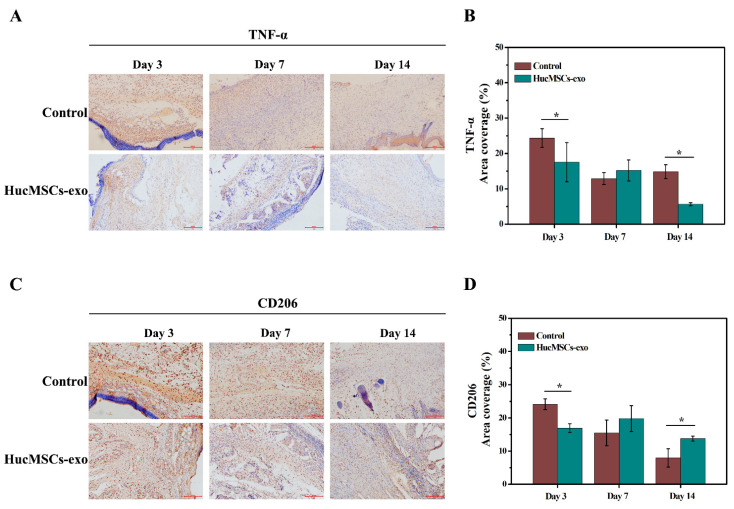
Immunohistochemistry analysis of TNF-α and CD206 expression in diabetic wound tissues on days 3, 7, and 14. Representative images of TNF-α (**A**) and CD206 (**C**) staining of wound tissues. The scale bar is 200 µm. Quantitative analysis of the positively stained area percentage of TNF-α (**B**) and CD206 (**D**). The data are expressed as mean ± SD (n = 3). * *p* < 0.05 compared with the control group.

**Figure 6 ijms-23-10421-f006:**
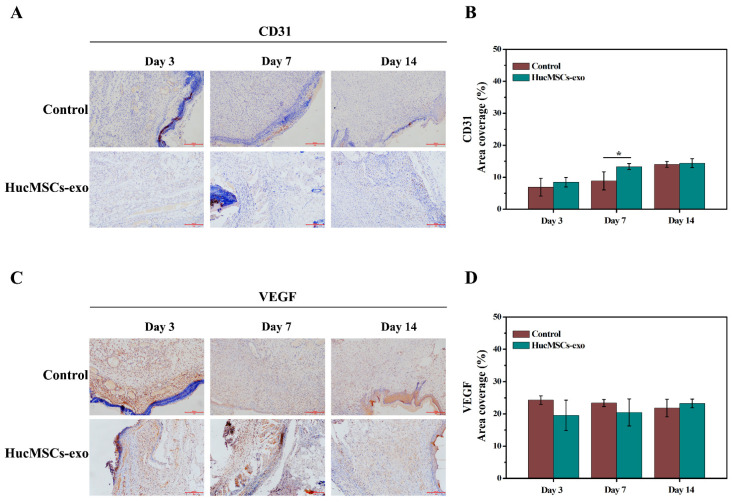
Immunohistochemistry analysis of CD31 and VEGF expression in diabetic wound tissues on days 3, 7, and 14. Representative images of CD31 (**A**) and VEGF (**C**) staining of wound tissues. The scale bar is 200 µm. Quantitative analysis of the positively stained area percentage of CD31 (**B**) and VEGF (**D**). The data are expressed as mean ± SD (n = 3). * *p* < 0.05 compared with the control group.

**Figure 7 ijms-23-10421-f007:**
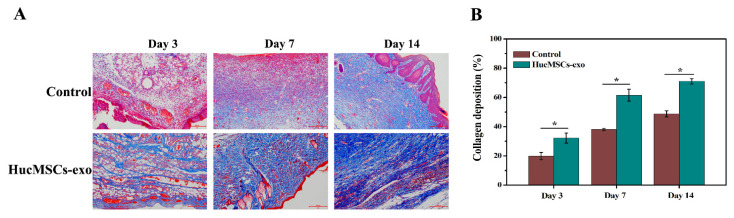
Masson’s trichrome staining in diabetic wound tissues on days 3, 7, and 14. (**A**) Representative images of Masson’s trichrome staining of wound sections. The scale bar is 200 µm. (**B**) Quantitative analysis of pigmentation areas (n = 3). * *p* < 0.05 compared with the control group.

## Data Availability

Not applicable.

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
