# Peer review of "Exosomes Derived from Human Umbilical Cord Mesenchymal Stem Cells Accelerate Diabetic Wound Healing via Promoting M2 Macrophage Polarization, Angiogenesis, and Collagen Deposition"

_ijms, 2022, doi:10.3390/ijms231810421_

Round 1
Reviewer 1 Report
This study demonstrates that exosomes derived from human umbilical cord mesenchymal stem cells accelerate diabetic wound healing in a rat model. 1. The abstract contains multiple non-standard abbreviations that are unexplained. Although many scientists are familiar with these abbreviations, this is not an acceptable practice and abbreviations must be explained at first use. 2. Line 16: TNF-α is not TNF-a. 3. Line 31: Please briefly discuss the classification of common types of chronic wounds. 4. Exosomes are a biological product with a risk of immune responses. The immunogenic potential of these exosomes should be discussed. It is a potential problem. 5. Streptozotocin-induced diabetes mellitus is a model of secondary diabetes. Moreover, streptozotocin may have effects that are independent of diabetes mellitus. This may not be the most ideal model. 6. Please explain all non-standard abbreviations in the text at first use. 7. Diabetic ulcers in humans are neuropathic ulcers (also known as neurotrophic ulcers). They are caused by neuropathy, an abnormal functioning in the nerves of the feet. This dimension of diabetic ulcers is absent in these animal models. 8. When the authors show quantitative data in the text or in the Figures, they should consistently provide the number of independent experiments in each group. 9. Is there any real perspective for clinical translation?Author Response
Please see the attachment.

Reviewer 2 Report
Long-term non-healing ulcerative skin defects caused by diabetes mellitus are one of the causes of disability and mortality. The search for alternative ways to accelerate the healing of skin ulcers, including diabetes mellitus, remains an urgent task to this day. As an alternative to cell therapy with mesenchymal stem cells, the possibility of using conditioned media from cells or a separate population of microvesicles - exosomes is considered. In this work, the authors clearly demonstrate the therapeutic potential of cord blood mesenchymal stem cell exosomes.
The disadvantages of the work include a typo on line 290, you should write pellet instead pallet. As well as, the lack of justification for the use of data representation in the group according to the Gauss-Poisson law. In the statistical processing section, this paragraph should clearly state which method for verifying the normal distribution of features was taken as the basis, preferably the w-Shapiro-Wilks test. This moment in the section of statistical data processing removes questions about the validity of the presentation of the obtained data in the text of the article.
